# Generalized Source-free Domain-adaptive Segmentation via Reliable Knowledge Propagation

## ABSTRACT

Unanticipated domain shifts can severely degrade model performance, prompting the need for model adaptation techniques (*i.e.*, Source-free Domain Adaptation (SFDA)) to adapt a model to new domains without accessing source data. However, existing SFDA methods often sacrifice source domain performance to improve adaptation on the target, limiting overall model capability. In this paper, we focus on a more challenging paradigm in semantic segmentation, Generalized SFDA (G-SFDA), aiming to achieve robust performance on both source and target domains. To achieve this, we propose a novel G-SFDA framework, Reliable Knowledge Propagation (RKP), for semantic segmentation tasks, which leverages the text-to-image diffusion model to propagate reliable semantic knowledge from the segmentation model. The key of RKP lies in aggregating the predicted reliable but scattered segments into a complete semantic layout and using them to activate the diffusion model for conditional generation. Subsequently, diverse images with multiple domain factors can be synthesized to retrain the segmentation model. This enables the segmentation model to learn domain-invariant knowledge across multiple domains, improving its adaptability to target domain, maintaining discriminability to source domain, and even handling unseen domains. Our model-agnostic RKP framework establishes new state-of-the-art across current SFDA segmentation benchmarks, significantly advancing various SFDA methods. The code will be open source.

## CCS CONCEPTS

• **Computing methodologies** → **Image segmentation**; • **Transfer learning** → *Source free domain adaptation.*

## KEYWORDS

source-free domain adaptation, semantic segmentation, diffusion model

## 1 INTRODUCTION

In real-world machine perception systems (e.g., autonomous driving [2]), unexpected domain changes in test distribution are commonly encountered, which can lead to a significant degradation in perception capability when applying pre-trained models [41, 46]. Therefore, the development of model adaptation methods is essential for enhancing the generalization capability of models in the

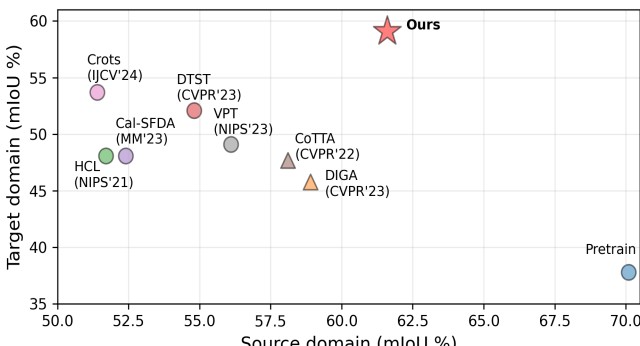

**Figure 1: The performance comparison of various methods on both the source and target domains in GTA → Cityscapes tasks under the source-free domain adaptation. All methods use the same pre-trained model.**

wild and improving the reliability of machine perception systems in dynamic environments[28]. To this end, Source-free Domain Adaptation (SFDA) has been proposed [25, 30], aiming to adapt a source-trained model to a target domain without accessing the source data. In particular, the SFDA methods for segmentation [23, 32, 50, 66] have been proposed to improve the models' adaptability of on target domain by self-training. However, as shown in Fig. 1, most SFDA methods sacrifice the performance of the source domain when improving target adaptability due to the catastrophic forgetting [22], which leads to the model's generalization capability remaining limited. In this paper, we focus on a more practical and challenging paradigm in semantic segmentation, Generalized SFDA (G-SFDA) [57], which requires the adapted model to perform well not only on source and target domains but even on unseen domains.

While the concept of G-SFDA [57] has been proposed in classification tasks for some time, there has been scarce exploration of G-SFDA in segmentation tasks. Relevant to this objective is the continual domain adaptation segmentation [46, 49], where techniques involve freezing some segmentation model's parameters to maintain its performance on the source domain. However, we find that while this approach indeed yields some benefits in maintaining performance on the source, its benefits on the target appear to be relatively limited compared to existing SFDA methods, as indicated by the triangular marker (CoTTA [46] and DIGA [49]) in Fig. 1. This is mainly because the adaptation for segmentation tasks is often complex and requires the model to update most or full parameters to enhance its adaptability to the target. This challenge poses difficulties for existing techniques in balancing adaptability to the target domain and discriminability to the source domain, and also limits its generalization to unseen domains.

In this paper, we introduce a novel idea from a generative perspective to address both target adaptation and source discrimination issues simultaneously: updating all parameters of the segmentation

model using synthetic multi-domain data to learn domain-invariant representations, as shown in Fig. 2. Based on this, we innovatively develop a G-SFDA framework, termed Reliable Knowledge Propagation (RKP), to propagate reliable semantic knowledge from the segmentation model through the text-to-image diffusion model [39]. The key of RKP is to aggregate the scattered reliable segments predicted by the segmentation model into a complete semantic layout and activate the text-to-image diffusion model for conditional generation. Subsequently, diverse training images with multiple domain factors (*e.g.*, *different weather or illumination environments*) can be synthesized through activated diffusion model. Under this drive, the segmentation model can jointly learn domain-invariant knowledge across multiple synthetic domains, enhancing its adaptability to target and empowering its discriminability to the source domain and even unseen domains.

Specifically, our RKP consists of three stages. *1) Reliable Knowledge Aggregation.* Given many reliable scattered segments from the pre-train segmentation model, activating the text-to-image diffusion model to synthesize target-specific data is difficult due to the lack of complete layout. One possible idea is to mix scattered segments into a complete semantic layout. The challenge here is to select appropriate mixing segments and determine where and how they should be mixed. To achieve this, we devise a Layout-Aware Mixing (LAM) technique, which adaptively mixes the scattered reliable segments into a complete layout in an optimizable manner. LAM retrieves the most appropriate segment for the unreliable region by matching the class distribution among reliable segment candidates and learns the most suitable affine transformation to determine the optimal mixing position for the selected segment. *2) Reliable Knowledge Injection.* With reliable LAM-driven target data, sufficient semantic-to-image pairings can be used to control the diffusion model. Taking mixed reliable semantic mask as a spatial condition and the caption of the category name component as the text prompt, the diffusion model can be effectively fine-tuned to the target domain. Observing noisy edges and semantics in spatial conditions, we propose uncertainty-guided fine-tuning for stabilizing the tuning process. *3) Reliable Knowledge Propagation and Learning.* By incorporating vocabulary representing various domain factors, *e.g.*, different simulation ('synthetic'/'real') or weather ('hot') or illumination ('night'), into the text prompts, the fine-tuned diffusion model synthesizes both target domain and out-of-domain data with various styles. Subsequently, these synthesized data are utilized to learn domain-invariant representations. With these efforts, RKP endows the segmentation model with adaptability to the target and discriminability to the source, as shown in Fig. 1. In summary, our contributions are as follows:

- For the first time, we introduce the text-to-image diffusion model in SFDA semantic segmentation task and develop a novel G-SFDA framework, called Reliable Knowledge Propagation (RKP). RKP can not only enhance adaptability to the target domain but also the discriminability of source domain and even unseen domains.
- In RKP, we devise a Layout-Aware Mixing (LAM) technique, that optimally mixes scattered reliable segments into complete layouts, aided by retrieving the most appropriate segments for uncertain regions through class distribution

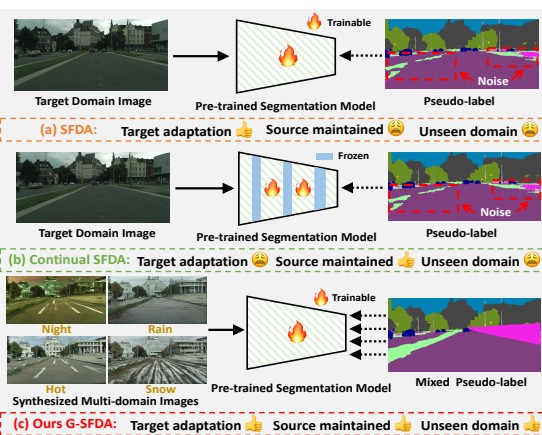

Figure 2: Comparison of our method with traditional SFDA methods and continual SFDA [46] in principle.

matching and determining the optimal mixing position of segments through affine transformation learning.
- With LAM-driven paired semantic-to-image data and corresponding text prompts, the diffusion model is fine-tuned to the target domain, while proposing uncertainty-guided fine-tuning to stabilize the fine-tuning by alleviating noise.
- Our framework RKP is model-agnostic and can be embedded into various SFDA methods, further boosting their generalization capabilities.

## 2 RELATED WORK

**Source-free Domain Adaptation (SFDA).** Previous SFDA methods in classification tasks propose various techniques such as distribution alignment[10, 29, 48], contrastive learning[65, 67], and model perturbation [21] to enhance adaptability without source data. However, these are challenging to apply in segmentation due to complex semantic features. SFDA in segmentation tasks often involves self-training [11, 19, 23, 47, 50] by filtering reliable pseudo-labels and retraining the model. Filtering methods include thresholds[60, 66, 68], adversarial training[58], and feature prototypes[6]. Anti-noise training methods [61] are also proposed for SFDA to reduce error accumulation by adding perturbations [26] or enhancing data [12, 32]. Despite progress in pseudo-label filtering and anti-noise training, overfitting to the pseudo-label noise of the target domain remains a challenge. Our method sidesteps high-noise pseudo-labels, leveraging reliable multi-domain samples to adapt to the source-trained model effectively, thus enhancing generalization and reducing error accumulation.

**Generalized Source-free Domain Adaptation (G-SFDA).** G-SFDA[57], an extension of SFDA, is expected to perform well across both source and target domains. While it's been mainly studied in classification tasks, addressing catastrophic forgetting remains a challenge. Existing methods preserve weights [3, 35] or update specific prompts [13, 43] to maintain history domain performance. However, these strategies are less effective in segmentation tasks due to complexity in balancing adaptability and discriminability across domains. In contrast, our novel G-SFDA method (RKP) synthesizes multi-domain data using a text-to-image diffusion model

**Figure 3: The pipeline of our RKP for G-SFDA: (a) Through layout-aware mixing, the prediction and uncertainty maps of target data from the segmentation model are exploited to generate mixed reliable samples with complete layouts. (b) Training ControlNet with mixed prediction as spatial conditions and captions composed of class names as text prompts to adapt the Diffusion model to specific target data. (c) Text prompts serve as style conditions, while mixed prediction serves as spatial conditions for controlling generation of multi-domain images, which can be used for training the segmentation model.**

and focuses on jointly learning domain-invariant representations, presenting a new idea tailored for segmentation tasks.

**Text-to-image Diffusion Model for Semantic Segmentation.** Some prior works explore the potential of pre-trained text-to-image diffusion model by synthesizing new training data [9, 34, 51, 56] for semantic segmentation tasks, offering a viable solution to alleviate the issue of data scarcity in semantic segmentation tasks. Nguyen *et al.* [34] and Wu *et al.* [52] optimize the cross-modal attention maps between text and images, considering them as semantic masks for image generation. Wu *et al.* [51] augment the diffusion model with perception heads and fine-tune the added units using a small number of target samples to generate paired data. However, such approaches are prone to generating domain-agnostic data with simple layouts, which impedes the model's ability to extract useful knowledge for domain adaptation tasks. In contrast to these methods, we focus on extracting reliable domain-specific knowledge from the source model to activate the diffusion model, thereby empowering it to generate diverse target-style training data.

## 3 METHOD

### 3.1 Problem Setting and Preliminary

**Problem Setting.** Given a pre-trained segmentation model $\mathcal{G}(\theta)$ trained on labeled source domain data $\mathcal{D}_s = \{(x_s^i, y_s^i)\}_{i=1}^{n_s}$, where $x_s^i$ is the source image and $y_s^i$ is the corresponding label, we aim to adapt the pre-trained model to target domain data $\mathcal{D}_t = \{(x_t^i)\}_{i=1}^{n_t}$ only consisting of $n_t$ unlabeled images, and the source domain data is not accessible. For traditional Source-Free Domain Adaptation (SFDA), the adapted segmentation model is only required to perform well on the target domain $\mathcal{D}_t$. In this paper, we consider

the Generalized SFDA (G-SFDA), where the adapted segmentation model needs to perform well on both domains $\mathcal{D}_s$ and $\mathcal{D}_t$.

**Preliminary.** Most works conduct self-training to optimize the $\mathcal{G}(\theta)$ as follows,

$$\arg\min_{\theta} \sum_{i}^{n_t} \sum_{l}^{H \times W} \mathcal{L}[\mathcal{G}(x_t^{(i,l)}|\theta), \hat{y}_t^{(i,l)}], \qquad (1)$$

where $\mathcal{L}$ is the cross-entropy loss and $\hat{y}_t^i$ is the pseudo-label. As previously mentioned, there are two issues with this approach. First, noise in pseudo-labels results in error accumulation during self-training, which reduces the adaptability to the target. Second, updating all parameters for specific target data reduces the discriminability of the model to the source, while updating some parameters will harm the adaptability to the target, which poses difficulties in balancing adaptability and discriminability. Next, we will introduce our solution that can better alleviate the above problems.

### 3.2 Overall

This paper develops a novel G-SFDA framework called Reliable Knowledge Propagation (RKP), which propagates knowledge from the segmentation model $\mathcal{G}(\theta)$ through the diffusion model and then facilitates $\mathcal{G}(\theta)$ to learn domain-invariant representations. Our main insight is to mix the scattered reliable segments predicted by $\mathcal{G}(\theta)$ into a complete layout, and then use them to guide diffusion model to learn specific target distribution, thereby generating unlimited and diverse annotated data. The synthesized data can be used to enhance the adaptability of $\mathcal{G}(\theta)$ to the target while maintaining the discriminability to the source.

**Figure 4: The pipeline of our Layout-Aware Mixing (LAM). LAM consists of two steps: (a) Given target image $x_t$, its pseudo-label $\hat{y}_t$ and uncertainty $u_t$, retrieving its most suitable reliable mixing segments $r_d$ through the label distribution of high uncertainty regions. (b) Mixing the retrieved segment $r_d$ and its corresponding image $x_{tr}$ with $\hat{y}_t$ and $x_t$ to make $r_d$ better cover the uncertain area in $u_t$ through an optimizable design.**

Our pipeline is shown in Fig. 3, including three stages: **a) Reliable Knowledge Aggregation (§3.3)** —mixing the scattered reliable segments into a complete layout by layout-aware mixing. **b) Reliable Knowledge Injection (§3.4)** —using mixed data to build spatial conditions and text prompts, and efficiently fine-tuning the diffusion model with controls. **c) Reliable Knowledge Propagation (§3.5) and Learning (§3.6)** —sampling multi-domain images under joint control of layout and style and leveraging synthetic images for learning domain-invariant representations.

### 3.3 Reliable Knowledge Aggregation

This section discusses two main aspects, extracting reliable segments and mixing them in a complete layout.

**Reliable Segments Extraction.** Simply, we utilize common Shannon entropy maps [45] as uncertainty measures, and then sort the predicted pixels for each class based on their entropy values and select the top-ranked segments as reliable ones. The details are as follows. Given the segmentation model $\mathcal{G}(\theta)$ and target domain data $x_t^i$, the $K$-dimensional soft-segmentation map can be predicted as $p^i = \mathcal{G}(x_t^i, \theta)$, where $p^i \in [0,1]^{H,W,K}$, $H$, $W$ is the height, width of $x_t^i$, $K$ is the number of classes. As $p^i$ can be interpreted as a discrete distribution over $K$ classes, thus the entropy map [45] of the prediction can be calculate as $u_t^i(x_t^i) = \frac{1}{\log(K)} \sum_{k=1}^{K} p^{i,k} \log p^{i,k}$. Given the pseudo-label $\hat{y}_t^i$ of $x_t^i$, we sort all the predicted pixels based on their entropy values and select the reliable ones,

$$\mathcal{R} = \text{Top}[\text{Sort}(\{\hat{y}_t^i\}_{i=1}^{n_t}, \{u_t^i(x_t^i)\}_{i=1}^{n_t}), \tau\%], \quad (2)$$

where $\text{Sort}(\cdot, \cdot)$ means sort the former list in descending order using the latter as metric, $\text{Top}[\cdot, \tau\%]$ means selecting top $\tau\%$ ranked pixels. Then, a connected region with the same semantic label on an image is regarded as a segment $r$. We denote these scattered reliable segments and their images as $\mathcal{R} = \{x_{tr}^i, r^i\}_{i=1}^{n_{tr}}$.

**Layout-Aware Mixing.** Given a target image $x_t^i$, its unreliable pseudo-label $\hat{y}_t^i$, its uncertainty map $u_t^i$, and a set $\mathcal{R}$ of reliable yet scattered images and segments, we aim to aggregate them into a complete layout. A naive way is using copy-paste [14, 44] to randomly paste images and segments in $\mathcal{R}$ onto the given $x_t^i$ and

$\hat{y}_t^i$. However, it always produces a messy image layout, making exploiting it difficult.

Then, the challenge here is how to select appropriate mixing objects and determine where and how they should be mixed. To solve this problem, we devise Layout-Aware Mixing (LAM), as shown in Fig. 4. The core idea is to cover the appropriate segments in $\mathcal{R}$ on the corresponding uncertain region in $\hat{y}_t^i$ to reduce the uncertain prediction in $\hat{y}_t^i$, thereby reasonably constructing a complete layout. LAM consists of two steps: LAM (1) first retrieves the most appropriate segment from $\mathcal{R}$ by matching the class distribution of the uncertain region and (2) then forms the mixing process as an optimizable problem.

Specifically, *(1) in the first step*, based on $\hat{y}_t^i$ and $u_t^i$, we analyze the connected regions in $\hat{y}_t^i$ and select the connected region $O$ with the highest uncertainty. We then calculate the pixel distribution within the maximum bounding rectangle of $O$. Despite significant semantic noise in $O$, the calculated pixel distribution can globally represent the semantic possibilities in this region. Thus, we retrieve the segment (denote as $r_d$) from $\mathcal{R}$ that most matches the class distribution of $O$ by follows,

$$r_d = \arg\max_r \text{SIM}[hist(r^i), hist(O)], r \in \mathcal{R}, \quad (3)$$

where $\text{SIM}[\cdot, \cdot]$ is the cosine similarity between two vectors, $hist(\cdot)$ is the calculated histogram of pixel numbers for the categories.

Next, *(2) in the second step*, $r_d$ will cover the uncertainty region $O$ in $\hat{y}_t^i$ with the most appropriate angle, scale and displacement. Inspired by the learnable spatial transformation [20], we devise an optimizable geometric (affine) transformation applied to $r_d$ so that the transformed $r_d$ can maximally coincide with the uncertainty region $O$. Let $\mathcal{T}_\omega$ be a 2D affine transformation with learnable parameters $\omega$. The pixels in $r_d$ are defined to lie on a regular grid $A[r_d] = \{(m, n)\}$, where $m$ and $n$ are the horizontal and vertical coordinates of arbitrary pixel in $r_d$, and its transformed pixels $\mathcal{T}_\omega(A[r_d])$ are also lie on a regular grid, *i.e.*,

$$\begin{pmatrix} m' \\ n' \end{pmatrix} = \begin{pmatrix} \omega_{11} & \omega_{12} & \omega_{13} \\ \omega_{21} & \omega_{22} & \omega_{23} \end{pmatrix} \begin{pmatrix} m \\ n \end{pmatrix}. \quad (4)$$

Let $\overline{r_d} \in \mathbb{R}^{H \times W}$ be the binary version of the segment $r_d$, where the segment region is 1 and the rest is 0. Then the optimization goal for $\omega$ is that, $\mathcal{T}_{\omega^\star} =$

$$\underset{\omega}{\arg\max} \frac{\sum[\overline{u}_t^i \cdot \mathcal{T}_\omega(A[\overline{r_d}]) \otimes \overline{r_d}]}{\sum[\mathcal{T}_\omega(A[\overline{r_d}]) \otimes \overline{r_d}] + \sum[\overline{u}_t^i] - \sum[\overline{u}_t^i \cdot \mathcal{T}_\omega(A[\overline{r_d}]) \otimes \overline{r_d}]}, \quad (5)$$

where $\otimes$ is the coordinate sampler, $\mathcal{T}_\omega(A[\overline{r_d}]) \otimes \overline{r_d}$ is the geometrically transformed segment, its sum represents the total area, and $\overline{u}_t^i \in \mathbb{R}^{H \times W}$ is to set all other values in $u_t^i$ to 0 except for the region $O$. The numerator represents the intersection of $\overline{r_d}$ and $O$, the denominator represents the union of $\overline{r_d}$ and $O$, and the entire formula represents the intersection and union ratio between the transformed segment $r_d$ and $O$. A larger value indicates that the transformed $r_d$ covers $O$ better, which also prevents the transformation scale of $r_d$ from being too large or too small.

To learn the transform parameters $\omega$, we introduce a small location network, *e.g*, Resnet-8 [16] with a regression layer. It takes the segment $r_d$ as input and outputs the affine transformation parameters. By optimizing the location network using Eq. (5), the optimal parameters $\tilde{\omega}$ can be obtained. With $\tilde{\omega}$, we denote the transformed $\overline{r_d}$ as a mask, *i.e.*, $\mathcal{M} = \mathcal{T}_{\tilde{\omega}}(A[\overline{r_d}]) \otimes \overline{r_d}$, the transformed image $\tilde{x}_{tr}^i = \mathcal{T}_{\tilde{\omega}}(A[\overline{r_d}]) \otimes x_{tr}^i$, the transformed segment as $\tilde{r}_d = \mathcal{T}_{\tilde{\omega}}(A[\overline{r_d}]) \otimes r_d$. Thus, the final layout-aware mixed image $x_{tm}^i$, mixed pseudo-label $\hat{y}_{tm}^i$, and mixed uncertainty $u_{tm}^i$ are obtained,

$$x_{tm}^i = x_t^i \cdot (1 - \mathcal{M}) + \tilde{x}_{tr}^i \cdot \mathcal{M}, \quad (6)$$

$$\hat{y}_{tm}^i = \hat{y}_t^i \cdot (1 - \mathcal{M}) + \tilde{r}_d, \quad (7)$$

$$u_{tm}^i = u_t^i \cdot (1 - \mathcal{M}). \quad (8)$$

Our LAM can be executed in multiple rounds to comprehensively cover semantic noise present in a given $\hat{y}_t^i$. See the Appendix for the specific algorithm flow. For the remaining edge noise in the mixed pseudo-label $\hat{y}_{tm}^i$, we will discuss it in the following sections.

## 3.4 Reliable Knowledge Injection

With reliable paired target data $\{x_{tm}^i, \hat{y}_{tm}^i, u_{tm}^i\}$, sufficient reliable data provide supporting material for controlling the diffusion model. Here, we adopt the ControlNet [63] to add the mixed reliable layout as spatial controls into the pre-trained stable diffusion [39, 42]. The details are as follows.

**Text Prompts.** We extract the class names present in $\hat{y}_{tm}^i$. Based on this list of class names, we can utilize the Large Language Models (LLM) such as ChatGPT [1] to generate sentences as text prompts, similar to the method described in [51].

**Spatial Condition.** Due to the efforts of LAM, the vast majority of $x_{tm}^i$ and $\hat{y}_{tm}^i$ are semantically consistent. Although noise at the class edges may still exist, it's possible to use $\hat{y}_{tm}^i$ as spatial conditions to control the main-body semantics of the generated image.

**Uncertainty Guided Fine-tuning.** One can aim at predicting the noise at each timestamp to fine-tune the diffusion model with ControlNet. However, as mentioned above, due to the noise at the edges, mismatches may exist between $x_{tm}^i$ and $\hat{y}_{tm}^i$ at the edges, resulting in noise in the data used for fine-tuning. Thus, we propose utilizing the mixed uncertainty to guide the training of noise predictions at each timestamp. Specifically, given an input image $x_{tm}^i$ and its corresponding uncertain map $u_{tm}^i$, the image diffusion algorithm

gradually adds noise $\epsilon$ to the image and generates a noisy image $z_t$, where $t$ denotes the number of times noise is added. Given time step $t$, text prompt $c_t$, and spatial conditions $c_f$ ($\hat{y}_{tm}^i$), we fine-tune the ControlNet to predict the noise in the confident regions (*i.e.* $1 - u_{tm}^i$) of $z_t$ during the reverse diffusion process as,

$$\mathcal{L}_{diff} = \mathbb{E}_{z_0, t, c_t, c_f, \epsilon \sim \mathcal{N}(0,1)} (1 - u_{tm}^i) \cdot \|\epsilon - \epsilon_\theta(z_t, t, c_t, c_f)\|^2. \quad (9)$$

Please refer to [63] for details on fine-tuning.

## 3.5 Reliable Knowledge Propagation

Through the fine-tuned ControlNet on the diffusion model, we can sample diverse data from a Gaussian distribution $\mathcal{N}(0, 1)$ with prompts, thus propagating the knowledge in the segmentation model to a wider target space and even a out-of-target space.

**Propagation to the Target.** On one hand, since ControlNet is trained with target data, we can use the original text prompts $c_t$ and spatial conditions $c_f$ to sample more target-style data $x_{syn}$. These sampled data can enhance the adaptation of the segmentation model to the target domain.

**Propagation to the Out-of-target.** On the other hand, inspired by some works on text-driven style transfer [27, 54], we can incorporate domain-specific factors (*e.g.*, the vocabulary of different weather, 'foggy', 'rainy') into the text prompt $c_t$ to generate out-of-target data $x_{dom}$ with various styles.

## 3.6 Reliable Knowledge Learning

With the mixed target data $\{x_{tm}^i, \hat{y}_{tm}^i, u_{tm}^i\}_{i=1}^{N_t}$, synthetic target-style data $\{x_{syn}^i, \hat{y}_{tm}^i, u_{tm}^i\}_{i=1}^{N_{syn}}$, and synthetic out-of-target data $\{[x_{dom}^{i,j}]_{j=1}^{N_d}, \hat{y}_{tm}^i, u_{tm}^i\}_{i=1}^{N_{dom}}$ ($N_d$ is the number of domain-specific text prompts.), we combine them to jointly learning domain-invariant representations for the segmentation model,

$$\mathcal{L}_{seg} = \frac{1}{N_t} \sum_i^{N_t} \mathcal{L}_w(x_{tm}^i, \hat{y}_{tm}^i, u_{tm}^i) + \frac{1}{N_{syn}} \sum_i^{N_{syn}} \mathcal{L}_w(x_{syn}^i, \hat{y}_{tm}^i, u_{tm}^i)$$

$$+ \frac{1}{N_{dom}} \lambda \sum_i^{N_{dom}} \sum_j^{N_d} \mathcal{L}_w(x_{dom}^{i,j}, \hat{y}_{tm}^i, u_{tm}^i), \quad (10)$$

where $\mathcal{L}_w(\cdot, \cdot, \cdot)$ is the cross-entropy loss with uncertain weighting [7, 36]. $\lambda$ is the weighting coefficient, which balances the adaptability to the target domain and the discriminability to the source and out-of-target domains. We use $\mathcal{L}_{seg}$ to optimize all parameters of the model $\mathcal{G}(\theta)$, unlike other works [35] only a part of the parameters, as segmentation tasks require a larger adjustment of optimization space to better adapt to both new and old domains.

## 4 EXPERIMENTS

### 4.1 Datasets and Experimental Setup

**Datasets.** We use a real-world dataset (Cityscapes [8]) alongside two synthetic datasets (GTA5 [38] and SYNTHIA [40]). Cityscapes dataset comprises 2,975 training and 500 validation images at a resolution of 2048 × 1024. GTA5 dataset comprises 24,966 images with a resolution of 1914 × 1052 and shares 19 categories with Cityscapes. SYNTHIA dataset comprises 9,400 images with a resolution of 1280 × 760 and shares 16 categories with Cityscapes.

| Method | Arch | road | sidewalk | Building | Wall | fence | pole | light | sign | vege. | terrain | sky | person | rider | car | truck | bus | train | mbike | bike | $mIoU_T$ | $mIoU_S$ | $mIoU_H$ |
|---|---|---|---|---|---|---|---|---|---|---|---|---|---|---|---|---|---|---|---|---|---|---|---|
| Generalized Source-free Domain Adaptation: GTA - Cityscapes (Val.) | | | | | | | | | | | | | | | | | | | | | | | |
| URMDA (CVPR'21) [11] | R | 92.3 | 55.2 | 81.6 | 30.8 | 18.8 | 37.1 | 17.7 | 12.1 | 84.2 | 35.9 | 83.8 | 57.7 | 24.1 | 81.7 | 27.5 | 44.3 | 6.9 | 24.1 | 40.4 | 45.1 | 51.6 | 48.1 |
| SFDA (CVPR'21) [30] | R | 91.7 | 52.7 | 82.2 | 28.7 | 20.3 | 36.5 | 30.6 | 23.6 | 81.7 | 35.6 | 84.8 | 59.5 | 22.6 | 83.4 | 29.6 | 32.4 | 11.8 | 23.8 | 39.6 | 45.8 | 54.6 | 49.8 |
| HCL (NIPS'21) [19] | R | 92.6 | 54.6 | 82.8 | 33.2 | 26.2 | 39.8 | 38.1 | 31.9 | 84.5 | 38.6 | 85.3 | 61.3 | 30.2 | 85.4 | 33.1 | 41.6 | 14.4 | 27.3 | 44.0 | 49.7 | 51.7 | 50.7 |
| SFDASEG (ICCV'21) [24] | R | 91.7 | 53.4 | 86.1 | 37.6 | 32.1 | 37.4 | 38.2 | 35.6 | 86.7 | 48.5 | 89.9 | 62.6 | 34.3 | 87.2 | 51.0 | 50.8 | 4.2 | 42.7 | 53.9 | 53.4 | 54.4 | 53.9 |
| DTST (CVPR'23) [66] | R | 93.5 | 57.6 | 84.7 | 36.5 | 25.2 | 33.4 | 44.7 | 36.7 | 86.8 | 42.8 | 81.3 | 62.3 | 37.2 | 88.1 | 48.7 | 50.6 | 35.5 | 48.3 | 59.1 | 55.4 | 54.8 | 55.1 |
| CROTS(IJCV'24) [32] | R | 92.0 | 52.4 | 85.9 | 37.3 | 35.8 | 34.6 | 42.2 | 38.4 | 86.9 | 45.6 | 91.1 | 65.1 | 36.1 | 87.3 | 41.6 | 51.1 | 0.0 | 41.4 | 56.2 | 53.7 | 52.4 | 53.1 |
| CrossMatch(ICCV'23) [59] | R* | 95.1 | 67.8 | 87.7 | 51.3 | 41.5 | 36.3 | 47.4 | 51.3 | 87.8 | 47.8 | 87.3 | 67.0 | 34.2 | 87.5 | 41.0 | 51.8 | 0.0 | 42.6 | 46.4 | 56.4 | - | - |
| RKP (Ours) | R | 95.9 | 62.2 | 86.9 | 39.6 | 36.9 | 38.7 | 44.4 | 49.0 | 89.6 | 46.7 | 90.8 | 66.0 | 41.4 | 90.3 | 56.0 | 45.3 | 34.5 | 50.5 | 62.1 | **59.3** | **61.4** | **60.3** |
| VPT(NIPS'23) [33] | M | 89.4 | 22.9 | 87.3 | 38.9 | 34.3 | 41.0 | 45.8 | 30.3 | 88.8 | 44.2 | 90.1 | 67.0 | 32.4 | 90.1 | 52.9 | 60.4 | 37.1 | 37.7 | 38.6 | 54.2 | 55.6 | 54.9 |
| RKP (Ours) | M | 93.5 | 25.8 | 89.3 | 40.2 | 36.6 | 41.2 | 49.3 | 34.7 | 93.4 | 44.6 | 94.5 | 71.2 | 37.4 | 91.1 | 54.4 | 65.7 | 51.7 | 44.8 | 42.6 | **58.0** | **58.4** | **58.2** |
| Generalized Source-free Domain Adaptation: SYNTHIA - Cityscapes (Val.) | | | | | | | | | | | | | | | | | | | | | | | |
| URMDA (CVPR'21) [11] | R | 59.3 | 24.6 | 77.0 | 14.0 | 1.8 | 31.5 | 18.3 | 32.0 | 83.1 | - | 80.4 | 46.3 | 17.8 | 76.7 | - | 17.0 | - | 18.5 | 34.6 | 39.6 | 43.7 | 41.6 |
| SFDA (CVPR'21) [30] | R | 67.8 | 31.9 | 77.1 | 8.3 | 1.1 | 35.9 | 21.2 | 26.7 | 79.8 | - | 79.4 | 58.8 | 27.3 | 80.4 | - | 25.3 | - | 19.5 | 37.4 | 42.4 | 44.5 | 43.4 |
| HCL (NIPS'21) [19] | R | 86.7 | 38.1 | 82.7 | 10.0 | 0.6 | 30.3 | 25.4 | 29.7 | 82.8 | - | 85.9 | 61.9 | 24.8 | 84.5 | - | 38.9 | - | 22.6 | 37.9 | 46.4 | 41.7 | 43.9 |
| SFDASEG (ICCV'21) [24] | R | 90.5 | 50.0 | 81.6 | 13.3 | 2.8 | 34.7 | 25.7 | 33.1 | 83.8 | - | 89.2 | 66.0 | 34.9 | 85.3 | - | 53.4 | - | 46.1 | 46.6 | 52.3 | 46.8 | 49.4 |
| DTST (CVPR'23) [66] | R | 88.9 | 45.8 | 83.3 | 13.7 | 0.8 | 32.7 | 31.6 | 20.8 | 85.7 | - | 82.5 | 64.4 | 27.8 | 88.1 | - | 50.9 | - | 37.6 | 57.3 | 50.7 | 42.6 | 46.3 |
| CROTS(IJCV'24) [32] | R | 89.4 | 41.6 | 82.7 | 15.1 | 1.2 | 34.7 | 33.7 | 25.7 | 85.7 | - | 87.9 | 66.6 | 34.6 | 85.4 | - | 45.9 | - | 43.5 | 49.6 | 51.3 | 45.1 | 48.0 |
| CrossMatch(ICCV'23) [59] | R* | 91.5 | 55.5 | 85.4 | 34.4 | 8.3 | 40.8 | 40.0 | 44.4 | 86.6 | - | 84.3 | 62.4 | 22.0 | 88.3 | - | 60.0 | - | 40.6 | 45.6 | 55.6 | - | - |
| RKP (Ours) | R | 91.6 | 51.0 | 85.2 | 39.0 | 25.2 | 31.6 | 41.0 | 45.4 | 89.0 | - | 88.9 | 63.6 | 33.5 | 88.9 | - | 37.9 | - | 49.1 | 60.9 | **57.8** | **49.9** | **53.6** |
| VPT(NIPS'23) [33] | M | 88.6 | 47.8 | 84.0 | 36.8 | 3.0 | 39.8 | 37.3 | 35.4 | 83.9 | - | 87.2 | 66.2 | 31.3 | 85.0 | - | 50.6 | - | 39.1 | 45.0 | 53.8 | 44.7 | 48.8 |
| RKP (Ours) | M | 89.5 | 50.6 | 87.4 | 40.5 | 3.6 | 40.8 | 41.0 | 36.8 | 87.4 | - | 88.6 | 68.6 | 33.1 | 87.2 | - | 50.7 | - | 42.9 | 48.9 | **56.1** | **48.9** | **52.3** |

**Table 1: Comparison of RKP with state-of-the-art methods on the generalized source-free domain adaptation for semantic segmentation. Segmentation architectures: R (DeepLabv2 ResNet-101), M (Segformer MiT-B5). $mIoU_T/mIoU_S$ reports the performance on the target/source domain. R* denotes CrossMatch [59] using two segmentation models with depth estimation.**

**Implementation Details.** We use the SegFormer [53] with MiT-B5 [16] and Swin-transformer[31], the Deeplab-v2 [4] with ResNet-101. For optimizer, learning rate, and configuration, we follow the same settings as described in methods [66] and [17]. The Stable diffusion V1.5 model [39] pre-trained on the LAION5B [42] dataset is used as our text-to-image diffusion model. The batch size is set to 4, and the model is trained for 20,000 iterations. The batch size for fine-tuning with diffusion using ControlNet is set to 2, with other parameters consistent with ControlNet settings. For all tasks, we train ControlNet for 50,000 iterations with image size $512 \times 512$, on an RTX 3090 GPU. In LAM, the localization network is ResNet-8 with global pooling [15] and an additional regression head [20]. LAM optimization is unsupervised, and training is stopped after 100 iterations. We utilize 8 fixed domain textual prompts for sampling multi-domain data, namely ['windy', 'hot', 'cold', 'hail', 'foggy', 'night', 'rainy']. For each sampling, we randomly select $N_d = 4$ domain text prompts from the list. For each adaptation task, a total of 10,000 multi-domain images are synthesized. The synthesized target-style images $N_{syn} = 2,000$, the synthesized out-of-target images $N_{dom} = 8,000$, totally 10,000 synthesized images. The trade-off hyper-parameter $\lambda$ is set to 0.1.

**Evaluation Metrics.** We use the mean Intersection over Union (mIoU) to measure performance on each domain and calculate the harmonic mean $mIoU_H = \frac{2*mIoU_S*mIoU_T}{mIoU_S+mIoU_T}$ between the source and target mIoU, and $mIoU_S$ and $mIoU_T$ are the mIoU on source and target data, respectively.

## 4.2 Comparison with SOTA Methods

We compare our method with the current state-of-the-art (SOTA) in two Source-Free Domain Adaptation (SFDA) scenarios: GTA5 → Cityscapes and SYNTHIA → Cityscapes, as shown in Table 1. In both benchmarks, our RKP significantly outperforms our baseline.

| | Generalization to Unseen Domains | | | | | |
|---|---|---|---|---|---|---|
| Method | Type | Labeled Seen | Citysacpes | BDD-100K | Map. | ACDC | Unseen Average |
| Source only | DG | ✓ | 36.8 | 32.4 | 31.4 | 24.2 | 29.3 |
| FSDR (CVPR'21) [18] | DG | ✓ | 44.8 | 41.2 | 43.4 | 24.8 | 36.5 |
| SAN-SAW (CVPR'22) [37] | DG | ✓ | 45.3 | 41.1 | 40.7 | 23.1 | 35.0 |
| SHADE (ECCV'22) [69] | DG | ✓ | 46.6 | 43.6 | 45.5 | 29.1 | 39.4 |
| CROTS [32] | SFDA | ✗ | 53.7 | 24.9 | 29.9 | 19.6 | 28.1 |
| DTST [66] | SFDA | ✗ | 55.4 | 27.5 | 23.7 | 17.1 | 23.4 |
| SFDASEG [24] | DG+SFDA | ✗ | 53.6 | 30.9 | 33.7 | 19.6 | 28.1 |
| DTST [66] + SHADE [69] | DG+SFDA | ✗ | 53.9 | 33.5 | 36.9 | 20.1 | 30.2 |
| RKP (Ours) | G-SFDA | ✗ | 59.3 | 41.7 | 41.6 | 26.6 | 36.6 |
| RKP (Ours) + SHADE [69] | G-SFDA | ✗ | 59.5 | **44.9** | **46.1** | **29.9** | **40.3** |

**Table 2: Comparison of model's generalization on unseen domains, BDD-100K, Mapillary (Map.), ACDC. DG means given labeled GTA data and SFDA means only given pre-trained segmentation model and unlabeled Citysacpes data.**

Also, in terms of performance both on the target and source domains, RKP establishes a new state-of-the-art in Generalized SFDA (G-SFDA). Besides, we conduct Unseen Domain Generalization (DG) experiments in Table 2 to validate the generalization capability of our method on unseen domains. On three challenging unseen domains, our approach significantly improves the model performance without accessing labeled source data. Moreover, using DG methods as pre-train, our approach even achieves superior performance on unseen domains compared to the state-of-the-art DG methods.

**GTA → Cityscapes.** When using ResNet-101 and MiT-B backbones, our method respectively improves upon the best published results by +2.9% and +3.8% on the target domain, and by +5.8% and +2.8% on the source domain. Overall, our method has increased the sota performance set in G-SFDA by 5.4% and 3.3%, respectively. In contrast, existing SFDA methods, *e.g.*, Crossmath[59] and VPT[33], while achieving notable performance gains on the target domain, often do so at the expense of the source domain and do not fully exploit the pre-trianed segmentation model's capabilities.

| EXP-ID | LAM | UFT | Syn-target | Syn-out-target | ST-Warm | $mIoU_T$ | $mIoU_S$ | $mIoU_H$ |
|---|---|---|---|---|---|---|---|---|
| #1 | | | | | | 36.8 | 71.9 | 48.7 |
| #2 | ✓ | | | | | 53.7 | 53.1 | 53.4 |
| #3 | ✓ | ✓ | ✓ | | | 56.1 | 53.9 | 55.0 |
| #4 | ✓ | ✓ | | ✓ | | 55.1 | 59.6 | 57.3 |
| #5 | ✓ | | ✓ | ✓ | | 54.9 | 58.7 | 56.7 |
| #6 | ✓ | ✓ | ✓ | ✓ | | 57.7 | 60.2 | 58.9 |
| #7 | | ✓ | ✓ | ✓ | ✓ | 56.6 | 59.4 | 58.0 |
| #8 | ✓ | ✓ | ✓ | ✓ | ✓ | **59.3** | **61.4** | **60.3** |

Table 3: Ablation studies on GTA → Cityscapes. $mIoU_T/mIoU_S$ is the model's performance on Cityscapes/GTA. LAM is Layout-Aware Mixing. UFT is Uncertainty Guided Fine-Tuning. Syn-target is synthesizing target-style images. Syn-out-target is synthesizing out-of-target images. ST-Warm is using the self-training-based method [66] as a warm-up.

**SYNTHIA → Cityscapes.** In this benchmark, the performance results are consistent with the previous scenario. Our method achieves state-of-the-art accuracy for both backbones ResNet-101 and MiT-B. Compared to the SOTA results, our method improves by 2.2% and 2.3% on the target domain and by 3.1% and 4.2% on the source domain using ResNet-101 and MiT-B backbones respectively.

**Generalization to Unseen Domains.** We test our method's generalization on unseen domains in Table 2. Existing SFDA methods struggle with unseen domains, even with top DG methods like SHADE as pre-training, failing to maintain model generalization. Our approach, using SHADE as pre-training, outperforms SFDA methods by 10.1% mIoU on average across three unseen domains. It even improves DG methods' performance on new domains by 0.9% mIoU. Despite only using models trained on the source (GTA) and unlabeled target (Cityscapes) data, our method matches SHADE's performance on three unseen domains.

## 4.3 Ablation Study and Analysis

To understand the effectiveness of our framework, we conduct the ablation study using the ResNet-101 backbone in the GTA5 → Cityscapes. We perform experiments with variations of each component, termed #1-#8, and report the results in Table 3.

In Table 3, the non-adapted source model (#1) achieves 71.9% mIoU on the source domain but only 36.8% mIoU on the target domain. Using LAM (#2) for reliable prediction and mixing, and retraining the source model using the mixed data, we improve mIoU on the target domain by 16.9%. However, LAM cannot maintain performance on the source domain, and the mIoU score of the source is reduced to 53.1%. Then, fine-tuning the diffuse model (UFT) and synthesizing target domain data (#3) brings a 2.4% mIoU gain in target performance and some source domain income with 0.8% mIoU. Further, generating multi-domain data for domain-invariant learning (#6), yields performance gains of 7.1% and 4.0% on the source and target domains, respectively. Although the state-of-the-art performance has been achieved at this time, the performance can be further enhanced by using the self-training-based method [66] as a warm-up to provide LAM with more reliable segment candidates (#8). Finally, the full RKP framework results in an average gain of 11.6% compared to the source model across both domains.

**Effectiveness of Layout-Aware Mixing (LAM).** The effectiveness of LAM can be inferred from two experiments: #1→#2 and

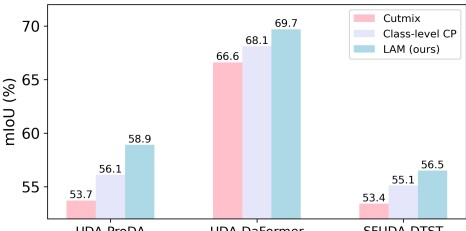

Figure 5: Comparison of our LAM with the existing CutMix [62] or Copy-paste (CP) [5, 14, 44] method on the GTA → Cityscapes task. We use ProDA [64] and DAFormer [17] as UDA baselines, and DTST [66] as the SFDA baseline.

#7→#8. Comparing #1 and #2, even without using the synthetic data, LAM still effectively improves the baseline method, which suggests that the mixed reliable data can independently work and benefit existing pseudo-label methods. For #7 and #8, removing LAM significantly reduces the gains obtained from synthesized data, resulting in a decrease of 2.0% and 2.7% on the source and target domains, respectively. This indicates that LAM effectively enhances the quality of synthesized data.

**Effectiveness of Uncertainty Guided Fine-Tuning (UFT).** The effectiveness of UFT can be validated from experiments #5 → #6. Using our UFT, the model achieve performance improvements of 1.5% and 2.8% on the source and target domains, respectively. This confirms the effectiveness of UFT in enhancing the gains from synthesized data.

**Effectiveness of Synthesized Multi-Domain Data.** The effectiveness of synthesized data can be validated from experiments #4→#6 and #3→#6. Adding synthesized target-style data in #4→#6 significantly improves the model's performance on the target domain, similar conclusions are drawn in #3→#6 for the source domain.

**Random CutMix/Copy-paste v.s. Layout-Aware Mixing.** Fig. 5 further verifies LAM by comparing it with some competitors. We upload traditional Unsupervised Domain Adaptation (UDA) methods as well as SFDA. We compare random CutMix [55, 62], random class-level Copy-paste [5, 14, 44], and our LAM on top of traditional UDA methods and SFDA methods. In both UDA and SFDA, our LAM achieves significant performance improvements compared to competitors in cross-domain segmentation using ResNet-101 and MiT-B5 as backbones. This indicates that our optimized way of constructing samples results in a more reasonable layout, while random pasting methods may disrupt the original layout structure, hence yielding more noticeable performance improvements.

**Combination with other SFDA Methods.** We combined our method with various types of SFDA methods, such as the initially poorly adapted method URMDA [11] and the better feature extraction network Swin-transformer [31], as shown in Fig. 8. After combining with these methods, our approach significantly improved their performance on both the source and target domains. This further demonstrates the compatibility of our method.

**Different Text Prompts Generators.** Table 4 compares fixed template prompts, like 'A photo of [class name],' to those generated by the Large Language Model (LLM) (e.g., ChatGPT) using class names. Results show fixed templates perform similarly in the target domain but degrade performance in the source domain. This may be due to their simplicity, causing overfitting during fine-tuning

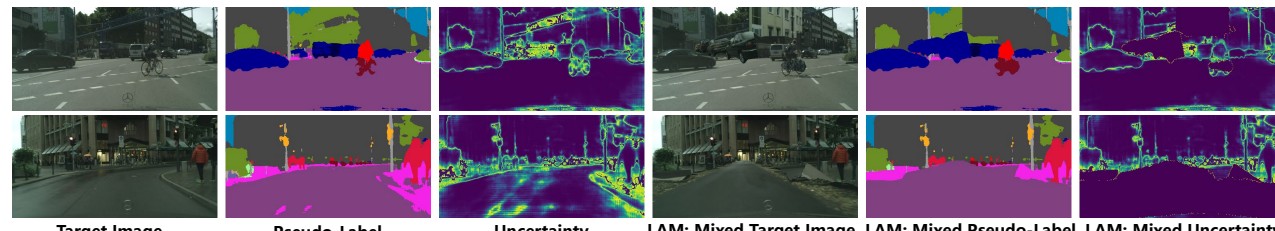

**Figure 6: High-quality mixed data from our LAM. The three columns on the left are predictions from the source model, and the three columns on the right are data mixed by our proposed LAM.**

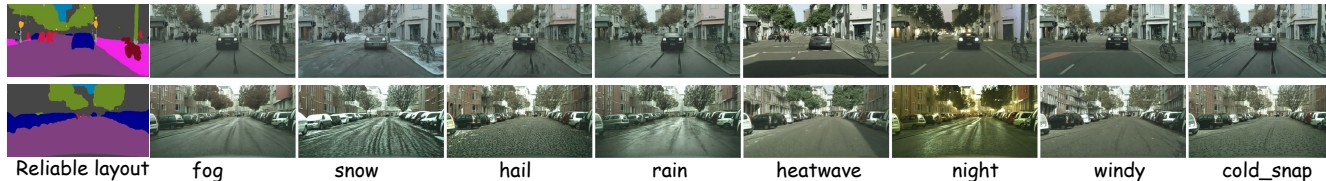

**Figure 7: High-quality multi-domain synthesized images. Sampling from our fine-tuned text-to-image diffusion model using the reliable layout as space conditions and text prompt as style condition.**

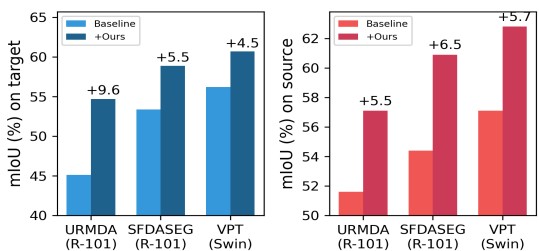

**Figure 8: The performance improvement of our method integrated into different SFDA baselines, URMDA [11], SFDASEG [24] and VPT [33]. Swin means using Swin-Transfomer [31] as the base segmentation model.**

| Method | $mIoU_T$ | $mIoU_S$ | $mIoU_H$ |
|---|---|---|---|
| 'A picture of [class name]' | 59.1 | 58.7 | 58.9 |
| LLM [1] | **59.3** | **61.4** | **60.3** |

**Table 4: Comparison of different text caption generators.**

of ControlNet and hindering the model's ability to learn strong generalization.

### 4.4 Qualitative Assessment

Fig. 6 shows that our LAM can blend discrete segments according to the semantics of the layout and effectively reduce prediction uncertainty. This provides data support for activating the text-to-image Diffusion model. Fig. 7 shows the high-quality multi-domain results synthesized by our method. It shows that our method effectively activates the diffusion model and empowers it with the ability to synthesize domain-specific data. Moreover, with the prompts of the domain text factor, realistic images conforming to the given semantic layout are still synthesized. More results can be found in the Appendix.

### 4.5 Hyper-parameter Analysis

We explore the impact of the number of synthesized instances $N_{syn}$ and domain text prompts $N_d$ in Fig. 9. The left figure shows that as

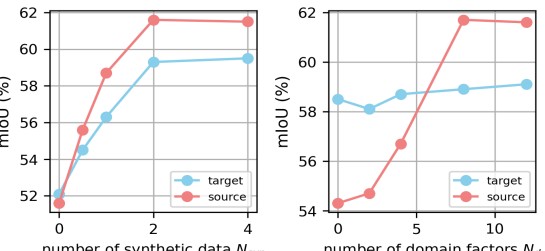

**Figure 9: The effect of the number of synthesized images and domain factors on the adaptability of G-SFDA.**

the number of synthesized data $N_{syn}$ increases, the model's performance gradually improves on both the source and target domains, stabilizing within a certain range. This illustrates that our synthetic target-style data can effectively improve the generalization ability of the model. The right figure shows that, with the increase in the number of domain text prompts $N_d$, the model's performance on the source domain gradually improves, indicating that the model gradually learns domain-invariant representations benefiting from our synthetic out-of-target data. Moreover, after reaching a certain range, the number of domain text prompts $N_d$ overall has little impact on the target domain.

### 5 CONCLUSION

In this paper, we focus on Generalized SFDA (G-SFDA) for semantic segmentation, aiming for robust performance across source and target domains. Our framework, Reliable Knowledge Propagation (RKP), leverages a text-to-image diffusion model to propagate reliable semantic knowledge from the segmentation model. By aggregating scattered reliable segments into complete layouts, RKP activates the diffusion model for conditional generation. This approach enables synthesis of diverse training images with multiple domain factors, enhancing source model adaptability across domains and achieving state-of-the-art performance on SFDA segmentation benchmarks. We hope that our work will bring new thinking to the community about model generalization.

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
