# OpenReview forum: "Generalized Source-free Domain-adaptive Segmentation via Reliable Knowledge Propagation"
_acmmm.org/ACMMM/2024/Conference — MM2024 Poster_

### Official Review · Reviewer_eWmk · 2024-05-19

**Rating:** 4
**Confidence:** 3

**Summary:**

In this paper, the author focuses on the semantic segmentation under the setting of GFSDA which requires high performance on both source and target domains.
The authors proposed Reliable Knowledge Propagation(RKP) to learn domain-invariant knowledge.
The method applies a data augmentation strategy and uses LLM, diffusion model conditioned on pseudo-label to generate synthesized samples with various domain labels.
The pipeline has three parts. The reliable knowledge aggregation mixes the scattered segments into a complete semantic layout using LAM. The reliable knowledge injection trains a ControlNet to adapt the diffusion model to specific target data. The reliable knowledge propagation and learning use the diffusion model to generate multi-domain images to learn domain-invariant knowledge.
Experiments show the framework establishes new SOTA across SFDA benchmarks.

**Strengths:**

1. Authors propose a pipeline to generate samples with multi-domain factors to increate the diversity of training samples.
2. Authors introduce LLM and stable diffusion to train a ControlNet to inject the reliable knowledge.
3. Experiments show good results on GSFDA benchmarks as well as unseen domains.

**Limitations:**

1. Authors need to show why the setting of GSFDA is important, although it has been applied to classification tasks.
Why the paper emphasis on the performance on source domains?
What practical scenarios does it apply to(line 93-94), and how does it differ from the settings of  SFDA and DG (I understand their definition)? An example of a real-world scenario would be helpful.

2. Is there any specific reason to use Resnet-8 in line 482 instead of commonly used one?

3. The pipeline use additional LLM and stable diffusion v1.5 model which are trained with large samples.
Do authors consider the computing and parameter overhead of this pipeline?
Do other methods in Table. 1 also use external models?
Authors show the improvement of different text caption generators in Table. 4. What about the diffusion model?
The authors need to further illustrate the relationship between the performance boost achieved by the method and the external models.

**Suitability:**

3

---

### Official Review · Reviewer_d6Sj · 2024-05-22

**Rating:** 4
**Confidence:** 4

**Summary:**

This paper introduces generalized source-free domain adaptation to semantic segmentation tasks, requiring the adapted model to perform well across source, target, and unseen domains. To achieve this, it primarily employs two strategies. First, Reliable Layout Matching (LAM) is introduced to exclude uncertain target predictions and paste the most similar class region onto that region using an optimizable geometric (affine) transformation. To encourage the model to learn reliable knowledge, this paper further utilizes target images and pseudo-labels after LAM. With the assistance of the diffusion model and GPT, it generates images of the same view across multiple domains, such as fog, rain, and snow. By learning with certain pseudo-labels and multiple generated domain data, the proposed method achieves state-of-the-art performance on the GTA5-to-Cityscapes and Synthia-to-Cityscapes benchmarks.

**Strengths:**

1. LAM is an interesting idea. It removes the uncertainties that the source model is not similar to. This allows a confident learning with pseudo-labels.
2. The diffusion model significantly allows the adapted model to perform well on target/unseen domains. The LAM strategy will also benefit source data performance as the paste regions are all source-reliable predictions, which all contribute to the generalized SFDA setting.
3. The proposed model achieves state-of-the-art performance across two well-known benchmarks.
4. The image/label/uncertainty visualization is good for understanding the ability of the LAM+diffusion model.
5. The architectural figure is clear and easy to understand.

**Limitations:**

1. In the main paper, the term "GTA5" is used correctly in some instances, while in others it is referred to as the "GTA dataset." This inconsistency should be addressed for clarity.
2. In lines 438-443, the explanation does not make sense to me because the uncertain region is identified by the source model as lacking confidence, which typically indicates either a significant domain gap or the presence of minority classes. Simply disregarding these uncertain areas could intuitively result in lower performance in these minority classes. I am also wondering why the learning process with disregard to minorities could surprisingly achieve good performance for minority classes.
3. When high-confidence target predictions contain noise, will this significantly impact the final model output, especially since these predictions are used for layout matching and image generation?
4. The idea appears to employ a combination of existing techniques, such as the diffusion model. However, LAM only achieves a 53.7 mIoU on the GTA5 to Cityscapes dataset. It would also be beneficial to know how LAM performs for each individual class within GTA5, particularly whether it results in notably poor performance for minority classes.
5. Given that the model design appears to significantly increase computational overhead, it would be beneficial to include a study on its efficiency, such as wall-clock time and Flops.
6. As the adaptation process is typically conducted on the target training set with final performance evaluated on the target validation set, it would be advantageous to also show visualizations for Cityscapes (Val) during evaluation. These should include scenarios with only LAM, with diffusion, and comparisons with ground truth and baselines.
7. It is also good to show how the number of generated domains will affect the performance of unseen domains.

**Suitability:**

3

---

### Official Review · Reviewer_zD1f · 2024-05-24

**Rating:** 4
**Confidence:** 3

**Summary:**

The paper introduces a framework for Generalized Source-Free Domain Adaptation (G-SFDA) in semantic segmentation, aimed at maintaining high performance across both source and target domains without source data access. The proposed method, called Reliable Knowledge Propagation (RKP), utilizes a text-to-image diffusion model to synthesize diverse training images that help the segmentation model learn domain-invariant knowledge. This method leverages segmented knowledge in the form of reliable but scattered segments, aggregates them into a comprehensive layout, and synthesizes new training data across multiple domains.

**Strengths:**

1. Novelty and Theoretical Approach: The paper introduces an innovative framework that combines reliable knowledge propagation with a diffusion model for domain adaptation in segmentation tasks, which is novel in the context of source-free domain adaptation (SFDA).
2.Comprehensive Evaluation: The paper includes extensive experiments and comparisons with state-of-the-art methods, providing a thorough validation of the framework’s effectiveness.The results show significant improvements, particularly in maintaining performance across both source and target domains.

**Limitations:**

1. Complexity and Scalability: The method's complexity, including the need for multiple components like diffusion model and LLM, may affect its scalability and ease of implementation in practical settings.
2. Dependence on High-Quality Segmentation: The effectiveness of the method heavily relies on the initial quality of segmentation predictions. If the segmentation model performs poorly, the subsequent steps may not significantly rectify these errors.
3. Distinction from Domain Generalization:While the G-SFDA method significantly advances the state-of-the-art in domain-adaptive segmentation by leveraging a novel reliable knowledge propagation framework, its conceptual overlap with domain generalization (DG) deserves acknowledgment. This paper could further differentiate G-SFDA from typical DG by discussing more distinct strategies used in DG that are not employed in G-SFDA.

**Suitability:**

3

---

### Official Review · Reviewer_BmEs · 2024-05-29

**Rating:** 4
**Confidence:** 3

**Summary:**

This paper focuses on a more practical and challenging paradigm (Generalized SFDA) in semantic segmentation, which aims to achieve robust performance on both source and target domains. The method aggregates the scattered reliable segments predicted by the segmentation model into a complete semantic layout, and exploits the text-to-image diffusion model for generating diverse training images with multiple domain factors to enhance generalization on source domain and target domain, even unseen domains.

**Strengths:**

1. The method proposes a new idea of aggregating the scattered reliable segments to generate target-style and out-of-target images for boosting adaptation ability.
2. An ablation study on each proposed component is shown.
3. The method outperforms existing baselines.

**Limitations:**

1. For reliable knowledge aggregation, since there are some classes are difficult to segment in the street scenes (such as small target classes),  whether the reliable set R are not include all classes?  How to deal with the classes that are not included?
2. The claim in line 533 and line 565 - line 568, the parameters that need to be optimized during the training process of framework include two parts: parameters of ControlNet on the diffusion model and parameters of the segmentation model. The proposed method has too many optimization parameters during the training process compared to other SOTA methods.
3. Whether the superiority over the current method is related to the amount of parameters it optimizes? If the proposed method can be trained with a comparable amount of parameters to other SFDA methods, it will be more helpful.
4. In line 771 to line 775 in ablation study, the experiment #7 removes LAM but uses synthesized data, so how is synthetic data generated?
5. Please discuss the limitations of the proposed approach.

**Suitability:**

3

---

### Meta-Review · Area_Chair_1Brs · 2024-07-02

**Recommendation:** Accept (Poster)
**Confidence:** 5

**Metareview:**

Pros:
- The paper introduces a novel framework that combines reliable knowledge propagation with a diffusion model for source-free domain adaptive segmentation.
- Extensive experiments and comparisons with state-of-the-art methods show significant improvements over baseline methods.
- The paper is well written and easy to follow

Cons:
- The method is complex and may involve significant computational overhead.
- The method relies on high quality initial segmentation predictions.